# Heterogeneity of synonymous substitution rates in the *Xenopus* frog genome

**Quintin Lau**[1]*, **Takeshi Igawa**[2], **Hajime Ogino**[2], **Yukako Katsura**[3], **Toshimichi Ikemura**[4], **Yoko Satta**[1]

**1** Department of Evolutionary Studies of Biosystems, Sokendai (The Graduate University for Advanced Studies), Hayama, Kanagawa, Japan, **2** Amphibian Research Center, Hiroshima University, Higashi-Hiroshima, Hiroshima, Japan, **3** Department of Cellular and Molecular Biology, Primate Research Institute, Kyoto University, Inuyama, Aichi, Japan, **4** Department of Bioscience, Nagahama Institute of Bio-Science and Technology, Nagahama, Japan

* quintin@soken.ac.jp

**Data Availability Statement:** All relevant data are within the paper and its Supporting Information files.

## Abstract

With the increasing availability of high quality genomic data, there is opportunity to deeply explore the genealogical relationships of different gene loci between closely related species. In this study, we utilized genomes of *Xenopus laevis* (XLA, a tetraploid species with (L) and (S) sub-genomes) and *X. tropicalis* (XTR, a diploid species) to investigate whether synonymous substitution rates among orthologous or homoeologous genes displayed any heterogeneity. From over 1500 orthologous/homoeologous genes collected, we calculated proportion of synonymous substitutions between genomes/sub-genomes ($k$) and found variation within and between chromosomes. Within most chromosomes, we identified higher $k$ with distance from the centromere, likely attributed to higher substitution rates and recombination in these regions. Using maximum likelihood methods, we identified further evidence supporting rate heterogeneity, and estimated species divergence times and ancestral population sizes. Estimated species divergence times (XLA.L-XLA.S: ~25.5 mya; XLA-XTR: ~33.0 mya) were slightly younger compared to a past study, attributed to consideration of population size in our study. Meanwhile, we found very large estimated population size in the ancestral populations of the two species ($N_A = 2.55 \times 10^6$). Local hybridization and population structure, which have not yet been well elucidated in frogs, may be a contributing factor to these possible large population sizes.

## Introduction

In comparisons between closely related species, it is well considered that different loci within a genome have different genealogical relationships. For example, many studies have compared several loci in humans with the three great ape species and showed that a majority (46.6–58.5% of loci) support the closest relationship between humans and chimpanzees; meanwhile, other loci support closer genealogical relationships between humans and gorillas or chimpanzees and gorillas [1–3]. Such variation in genealogical relationships between different gene loci can be attributed to large ancestral population size and short duration of successive speciation; this large ancestral population size can sometimes be resulting from heterogeneity of mutation

**Funding:** QL was supported by Goho Life Sciences International Fund.

**Competing interests:** The authors have declared that no competing interests exist.

rates among loci. With the increase in available genomic data, there is potential to explore these genealogical relationships in any species groups, with particular reference to investigating heterogeneity of mutation rates.

Such inconsistency of species genealogy with gene genealogy among loci has also been explored in anuran frogs, where different genealogies were explored among four lineages: *Silurana*, *Xenopus*, *Pipa*, and *Hymenochirus* [4]. It is generally accepted that *Silurana* and *Xenopus* form a clade, and this genealogy was confirmed to have the strongest statistical support following Bayesian analysis [4]. Indeed, revalidation of *Xenopus* topology and taxonomy [5] classifies *Silurana* as a subgenus within *Xenopus*, which includes *X. tropicalis*. Based on recently published genomic data, the common ancestor of *X. laevis* and *X. tropicalis* is estimated to have diverged ~48 mya [6]. In addition, *X. laevis* is a tetraploid species, comprised of (L) and (S) sub-genomes that diverged ~34 mya and hybridized ~17 mya, while *X. tropicalis* remained a diploid species [6]. We are interested in how this kind of species history is reflected in the genome, especially through demographic history of these species. However, to accurately know the demographic history of these species, we must first ascertain the extent of mutation rate heterogeneity.

One of the first studies to investigate synonymous substitution rates (equivalent to mutation rate, according to neutral theory) in amphibian genes was based on singular nuclear DNA sequences [7], including *c-myc*, *slug*, and tyrosinase precursor gene, in different anuran lineages (summarized in Table 1). Using more comprehensive genomic sequence data, a more accurate average synonymous substitution rate among over 8000 orthologous genes in *X. laevis* and *X. tropicalis* (identified using BLASTP followed by synteny agreement using BAC-FISH) was reported to be 3.0 to 3.2 $\times 10^{-9}$ substitutions/site/year [6]. However, further details have yet to be reported, in relation to the range and variation of synonymous divergence. Thus, in this study we further examine the publically available *Xenopus* genomes to investigate whether there is synonymous substitution rate heterogeneity in specific genomic regions or chromosomes.

## Materials and methods

### Data collection and calculation of synonymous substitutions

Three *Xenopus* genomes/sub-genomes were downloaded from Xenbase (http://www.xenbase.org/, RRID:SCR_003280): (i) XTR: *Xenopus tropicalis* (v9.1 genome assembly), (ii) XLA.L: *Xenopus laevis* (v9.1 genome assembly) L sub-genome, and (iii) XLA.S: *X. laevis* S sub-genome. Genomes were aligned using Synteny Mapping and Analysis Program (SyMAP) v 4.2 [8] and genomic locations for all orthologs between genomes of the two species or homoeologs between sub-genomes were compiled; the full data is available on the XenOrtho database

**Table 1. Summary of synonymous substitution rates estimated in Anuran frogs in previous studies.**

| Synonymous substitution rate (x10$^{-9}$ substitutions/site/year) | Species | Gene (s) | Nucleotide length | Reference |
|---|---|---|---|---|
| 0.92–1.53 | Three lineages of *Eleutherodactylus* | *c-myc* | ~1340 bp | [7] |
| 1.03 (0.68–1.42) | *Xenopus tropicalis** –*Xenopus laevis* | *slug* | 798 bp | [7] |
| 1.69 (1.14–2.45) | *Boophis xerophilus*–*Micrixalus fuscus* | *tyrosinase precursor* gene exon 1 | ~500 bp | [7] |
| 3.35 (2.43–4.52) | *Aglyptodactylus madagascariensis*–*Fejervarya syhadrensis* | *tyrosinase precursor* gene exon 1 | ~500 bp | [7] |
| 3.0† | *X. tropicalis*—*X. laevis* | 8806 homoeologous genes from genome | unspecified | [6] |
| 3.2# | *X. laevis* L–*X. laevis* S | | | |

*Silurana tropicalis* now referred to as *Xenopus tropicalis*; based on Ks/2T whereby T is 48 mya† and 34# mya for divergence between *X. laevis*- *X. tropicalis* and *X. laevis* L–*X. laevis* S, respectively.

(https://sites.google.com/view/xenorthodb/xenortho-db). A total of 1742 orthologous/homoeologous genes that had annotations in all three genomes/sub-genomes were collated, and we then excluded genes with overlapping annotations and genes with orthologs or homoeologs across different chromosomes; finally we obtained a set of 1598 orthologous loci from the three entire genomes and sub-genomes (54–4509 bp per gene, total 875911 bp).

Geneious® 11.1.5 (https://www.geneious.com) was used to extract sequences for the 1598 orthologous/homoeologous genes, using annotated gene names as search queries. Next, coding sequences (CDS) from the three genomes were aligned and gaps were excluded. For each genome pair combination (XTR-XLA.L, XTR-XLA.S, and XLA.L-XLA.S), we calculated the number ($d$) and proportion ($k = d/L$) of synonymous substitutions, whereby total length ($L$) of synonymous sites was calculated using MEGA-X-CC [9] with Nei-Gojobori method and Jukes Cantor correction for multiple hits. These parameters ($d$ and $k$) were calculated and collated separately for each chromosome.

## Maximum likelihood estimates of X and Y

Using the number of synonymous substitutions ($d$) and length ($L$), we utilized a maximum likelihood method that takes into account rate heterogeneity ($\alpha$) across loci and is based on a discrete gamma model by Yang [10]. $\alpha < 1$ is indicative of high heterogeneity, while infinitely large $\alpha$ denotes constant substitution rate or no heterogeneity. Before ML estimations, we first confirmed that $d$ fits with a negative binomial distribution using fitdistrplus package implemented in R [11]. Then, we calculated maximum likelihood (ML) estimates of $\alpha$ with assumption of $X = 0$; then $\alpha$, $d$, and $L$ were used for ML estimates of $X = 4N_Ag\mu$ and $Y = 2\mu t$, whereby $N_A$ represents ancestral population size, $t$ represents divergence time of two species compared, $g$ represents generation time, and $\mu$ represents mutation rate (synonymous substitution rate) per site per year. We calculated $X$ and $Y$ for two $\alpha$ values: infinitely large $\alpha$, and $\alpha$ values that gave best ML estimates of $X$ and $Y$. This was conducted independently for each chromosome, and also for all chromosomes collated together ('genome-wide'). We then calculated $N_A$ and $t$ assuming a generation time of 1 year and $\mu = 2.05 \times 10^{-9}$ substitutions/site/year; this synonymous substitution rate is calculated based on data by Session et al. [6] ($T^* = 0.0308 = T \times \mu$, where $T$ is ~15 mya, the time of pseudogenisation = assumed time of allotetraploidization, and $\mu$ is synonymous substitution rate).

## Statistical analyses

All statistical tests were conducted in R [11] or GraphPad Prism version 8.1.1 (GraphPad Software, La Jolla, CA, USA; www.graphpad.com). Within each of the three pairwise comparisons (XTR-XLA.L, XTR-XLA.S, and XLA.L-XLA.S), we used Analysis of Variance (ANOVA) test in R to determine if there are differences in proportion of synonymous substitutions ($k$) between chromosomes; Tukey multiple comparisons of means (95% confidence interval) was also implemented in R to identify which specific chromosomes are different to each other. To analyze whether there are changes in $k$ (*or K*) within each chromosome, we performed the partial *F*-test using ANOVA command in R to compare linear [lm($W$~$Z$)] and quadratic [lm($W$~$Z$+ I ($Z$^2)] models of $k$ (represented by $Z$) in relation to chromosome position ($W$).

## Results and discussion

### Synonymous divergence between and within chromosomes

We have utilized publically available genomic data from *X. laevis* and *X. tropicalis* to identify heterogeneity in the proportion of synonymous substitutions ($k$). Genome-wide mean $k$

between genes was lower in the pairwise comparison of XLA.L and XLA.S sub-genomes ($k = 0.183 \pm 0.043$, median = 0.184) compared to inter-species comparisons ($k = 0.227 \pm 0.055$ or $0.235 \pm 0.056$, median = 0.224 or 0.232) (Table 2). This is expected, since *X. tropicalis* and *X. laevis* diverged ~48 mya whereas the two *X. laevis* sub-genomes diverged ~34 mya [6], and this additional time allowed for increased accumulation of synonymous substitutions. Overall, the $k$ values in our study are slightly lower than those previously estimated from genomic data (median XTR-XLT $Ks$ = 0.286, XLA.L-XLA.S $Ks$ = 0.218) [6]. This might be attributed to the different approaches in extraction of orthologous/homoeologous genes; our approach utilized alignment of synteny blocks, which could be more conservative and focused on single copy orthologs whilst avoiding paralogs and orthologs located on different chromosomes.

When we investigated $k$ between each chromosome (i.e. at the inter-chromosomal level), there were no significant differences when comparing the *X.laevis* sub-genomes (Table 2, Fig 1A). However, when comparing either XLA.L or XLA.S with XTR, we found that $k$ was higher in orthologs located in chromosome 10 (XTR10) relative to all other chromosomes ($p < 0.0001$, Table 2, Fig 1B and 1C). In addition, orthologs had higher divergence in chromosome 5 compared to chromosome 3 ($p < 0.001$, Table 2, Fig 1B and 1C). The significantly higher $k$ in XTR10 when comparing XLA to XTR could be related to the dynamic fusion of chromosomes 9 and 10 in *X. laevis* [12], which we discuss later in this section.

Next, we examined intra-chromosomal divergence between XLA.L-XLA.S and XLA-XTR and found variation in $k$ between certain chromosomes. In chromosomes 1–7, a non-linear relationship, whereby $k$ increases with distance from the centromere (Fig 2A and S1 Fig), was confirmed by partial F-test, irrespective of the sub-genome (XLA.S, XLA.L, or XTR) used for chromosome location ($Pr(F) < 0.001$, S1 Table). This intra-chromosomal heterogeneity suggests that recombination is more prominent with distance from the centromere, and supports the finding that recombinations and local synonymous substitutions are positively correlated [13, 14]. Furthermore, when we investigated CG composition in full genome data at each chromosome, we found higher CG composition towards the telomeres (S2 Fig). This tendency of higher CG (that is, potentially higher CpG methylation) may explain the higher synonymous substitution rates away from the centromere. While an association between GC content and recombination rate has not been demonstrated in amphibians to date, a positive correlation has been identified in many other species [15–17]. The presence of transposable elements (TEs) may also contribute to heterogeneity in substitution rates within a chromosome. In

**Table 2. Proportion of synonymous substitutions ($k = d/L$) calculated from pairwise comparisons of the three *Xenopus* genomes/sub-genomes.**

| Chromosome | # genes | Proportion of synonymous substitutions, $k = d/L$[mean±s.d.] | | |
|---|---|---|---|---|
| | | XLA.L-XLA.S | XTR-XLA.L | XTR-XLA.S |
| Genome-wide | 1596 | $0.183 \pm 0.043$ | $0.227 \pm 0.055$ | $0.235 \pm 0.056$ |
| 1 | 299 | $0.179 \pm 0.043$ | $0.218 \pm 0.051$ | $0.228 \pm 0.053$ |
| 2 | 187 | $0.184 \pm 0.041$ | $0.224 \pm 0.049$ | $0.232 \pm 0.048$ |
| 3 | 206 | $0.181 \pm 0.037$ | $0.217 \pm 0.046$ | $0.219 \pm 0.048$ |
| 4 | 185 | $0.181 \pm 0.041$ | $0.222 \pm 0.056$ | $0.232 \pm 0.058$ |
| 5 | 164 | $0.184 \pm 0.044$ | $0.238 \pm 0.061$ | $0.244 \pm 0.056$ |
| 6 | 148 | $0.178 \pm 0.043$ | $0.230 \pm 0.056$ | $0.235 \pm 0.057$ |
| 7 | 109 | $0.185 \pm 0.043$ | $0.233 \pm 0.053$ | $0.238 \pm 0.051$ |
| 8 | 122 | $0.190 \pm 0.047$ | $0.223 \pm 0.059$ | $0.237 \pm 0.061$ |
| 9 | 102 | - | $0.230 \pm 0.058$ | $0.239 \pm 0.062$ |
| 10 | 74 | - | $0.276 \pm 0.053$ | $0.279 \pm 0.056$ |
| 9_10 | 196 | $0.190 \pm 0.050$ | - | - |

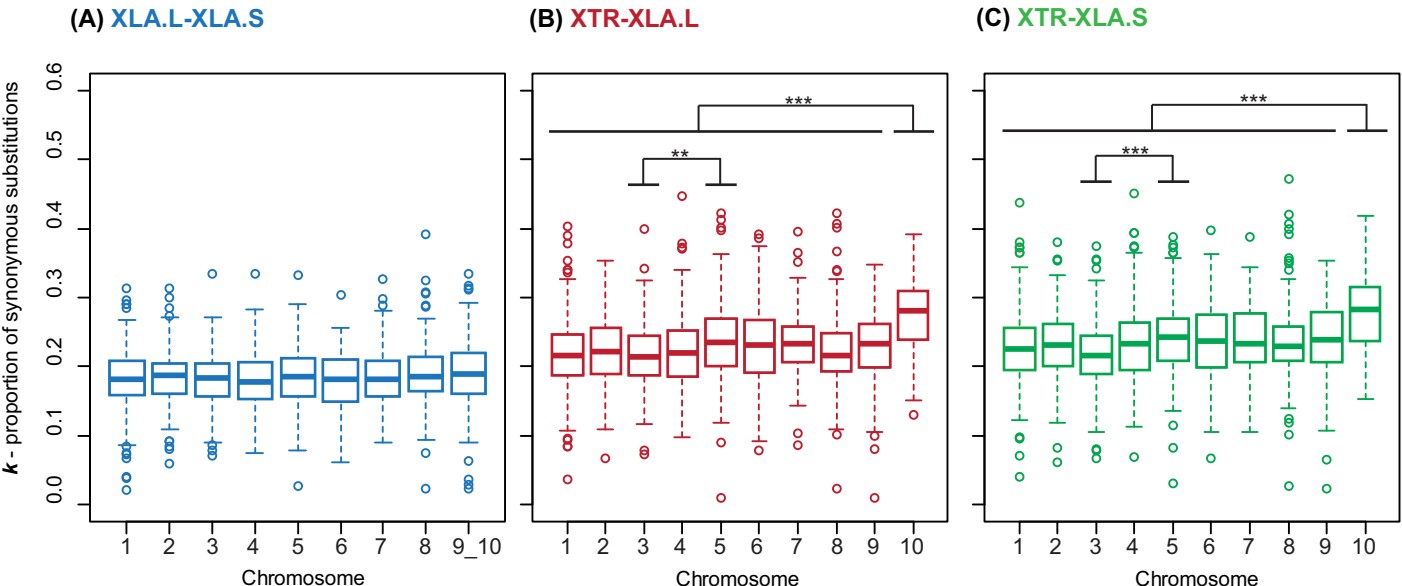

**Fig 1. Boxplots depicting proportion of synonymous substitutions (*k*) for each chromosome.** Between (A) *X. laevis* L (XLA.L) and S (XLA.S) sub-genomes, (B) *X. tropicalis* (XTR) and XLA.L, and (C) XTR and XLA.S. **p < 0.01, ***p < 0.0001 (adjusted p-values from Tukey multiple comparisons of means).

*Drosophila melanogaster*, TEs tended to be accumulated in the proximal region (i.e. near centromere) of autosomes [18]; transposon density was also found to be negatively correlated with recombination rate in this species [19]. Moreover, a study in *X. tropicalis* found that frequency of TEs was negatively correlated with GC content [20], but chromosomal location of only specific transposons have been mapped in *X. laevis* [6]. Based on these previous studies, there may be a lower frequency of TEs in the distal parts of the *Xenopus* chromosomes along with high CG content, which may be related to high recombination rates.

Exceptional cases were observed in chromosomes 8, 9, and 10. In chromosome 8, *k* varied depending on the genome used as reference for location. When we used XTR or XLA.L genomes as references, the relation between *k* and location was non-linear (Fig 2B, *Pr(F)* < 0.01), much like in chromosomes 1–7. However, when XLA.S was used, there was no bi-directional increase in *k* away from the centromere (Fig 2C, *Pr(F)* = 0.08–0.293). This is likely attributed to the chromosomal rearrangements and increased deletions previously reported in the S genome [6]; specifically, there is an inversion in the p-arm of XLA8S as well as homoeologous identity between the XLA8S p-arm and XLA8L q-arm (S3A Fig). We note that XLA3S also has chromosomal rearrangements but instead has a non-linear relation between *k* and location; this may be attributed to a lower degree of rearrangement and/or inversion being mainly limited to the q-arm of XLA3 (S3B Fig).

XTR-XLA and XLA.L-XLA.S intra-chromosomal examination of chromosomes 9 and 10 revealed contrasting relationships between *k* and chromosome location. The relationship between *k* and location was opposite between XTR9 and XTR10: *k* (XTR-XLA) increased with location in XTR9 (Fig 2D) but decreased with location in XTR10 (Fig 2E). Within the homoeologous XLA9_10, higher *k* in all pairwise genome comparisons was observed in the p-arm (Fig 2F and S1 Fig), which is predominantly equivalent to XTR10 based on cytogenetic mapping of chromosomes (S3C Fig); *k* then decreases distally in the q-arm of XLA9_10, which is mostly orthologous to XTR9 [6]. Chromosomes 9 and 10 probably fused sometime between the divergence of *X. laevis* with *X. tropicalis* (~48 mya) and the divergence of *Xenopus* (L) with *Xenopus*

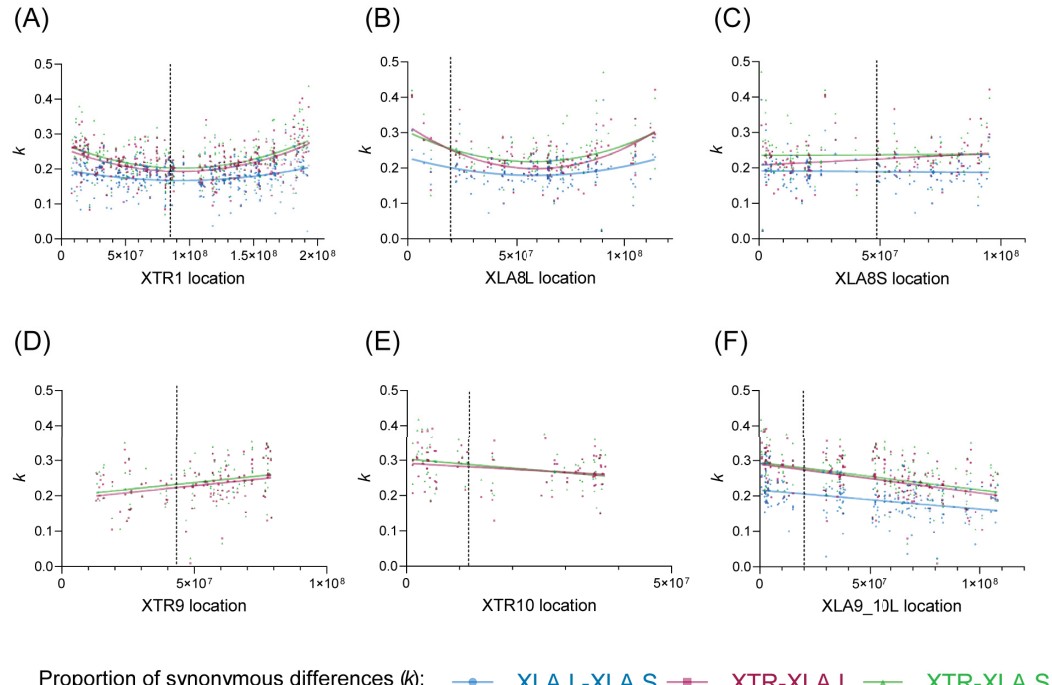

**Fig 2. Relationship patterns between proportion of synonymous substitutions (*k*) and chromosome location.**
Representative figures displaying different relationship patterns for each genome pair combination (XTR-XLA.L in red, XTR-XLA.S in green, and XLA.L-XLA.S in blue). (A) Chromosome 1 (e.g. XTR1) showed a common non-linear relationship between *k* and location. This higher *k* in distal parts of the chromosome was also seen in chromosomes 2 to 7, irrespective of the genome used for location in the *x-axis* (S1 Fig). (B, C) Chromosome 8 showed contrasting patterns based on the genome used for location reference: (B) XTR8 (S1 Fig) and XLA8L showed higher *k* in the distal part of chromosome, while (C) XLA8S had no marked relationship between *k* and location, likely due to intra-chromosomal rearrangements [6]. (D) In XTR9, *k* increased distally with location, while (E) it decreased in XTR10; (F) in the homoeologous XLA9_10, *k* decreased with location. Centromere positions are indicated by dotted vertical line and estimated based on position of frog centromeric repeat 1 (Fcr1) [21] or centromeric markers from *X. tropicalis* [22]. Full results are shown in S1 Fig.

(S) (~34 mya). It seems possible that homoeologous XLA9_10 chromosome, when compared to chromosomes 1–8, had a shorter amount of time for intrachromosomal recombination or accumulation of substitutions away from the centromere. The fusion of chromosomes 9 and 10 in XLA may have also impacted on mutation and recombination within XLA9_10. In addition, it may also be possible that different chromosomes experienced high recombination and mutation rates, while others could be under greater evolutionary constraint. For example in birds, synonymous substitution rates were higher in microchromosomes compared to macrochromosomes [23]. While *Xenopus* frogs do not have microchromosomes, our finding of a higher proportion of synonymous substitutions in XTR10 (Fig 1) could support that recombination rates differ between and within anuran chromosomes.

## Estimation of ancestral population size and species divergence time

Among all pairwise genome comparisons across the ten chromosomes, maximum likelihood estimates of $\alpha$ ranged from $\alpha = 14.9–41.8$ (S2 Table) with an assumption of no ancestral polymorphism. Using these initial $\alpha$ values, we estimated $X$ and $Y$ again and sometimes found that a smaller or larger $\alpha$ gave better ML estimates (S2 Table); these were subsequently used. ML estimates of $X = 4N\mu$ and $Y = 2\mu t$ were significantly different ($p < 0.001$) when using an infinitely large $\alpha$ (i.e. no rate heterogeneity) compared to using $\alpha$ with best ML estimates (Fig 3);

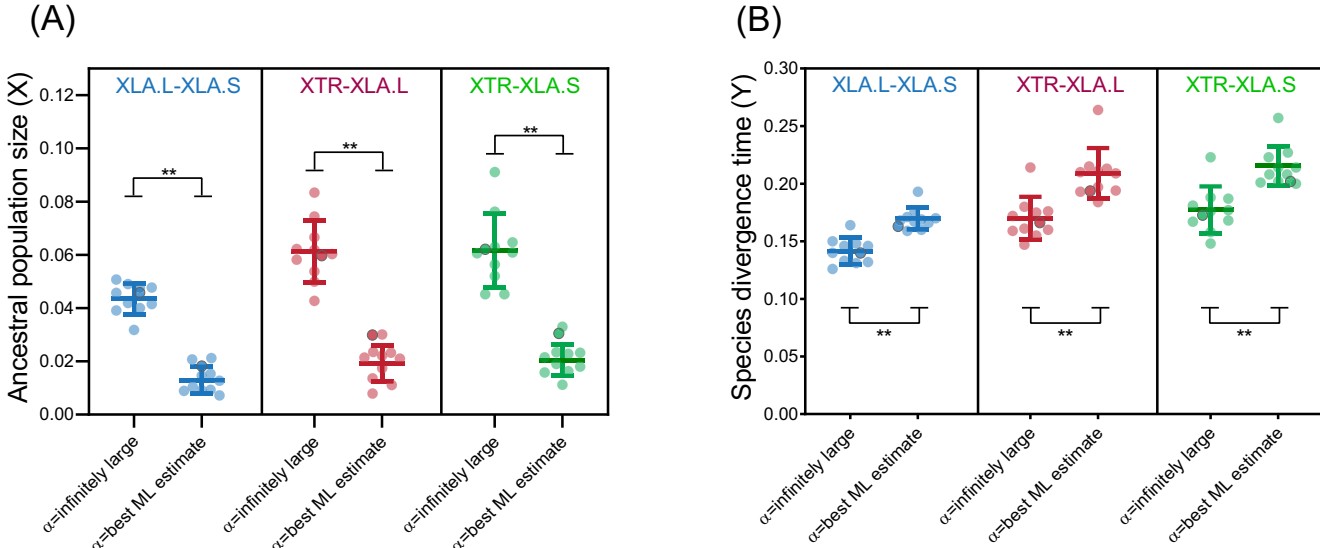

**Fig 3. Best maximum likelihood estimates of $X = 4N\mu$ and $Y = 2\mu t$.** Estimates were used to calculate ancestral population size and species divergence time, respectively (in ancestor of XLA-XTR or XLA.S-XLA.L). These estimates were significantly different compared to those using infinitely large α (**p < 0.0001, t-test). Line bars represent mean ± s.d.; each point represents each chromosome, with genome-wide results indicated by dark circles.

this provides evidence that some rate heterogeneity is present in the *Xenopus* genome. Using the α values with best ML estimates, we calculated species divergence time and ancestral population size.

Firstly, chi-square tests for all chromosomes showed no significant differences between observed synonymous divergence against a negative binomial distribution (all p > 0.01; S3 Table). The only exception was when data from all chromosomes were compiled (chi-square p-value < 0.01), likely due to intra-chromosomal variation, and thus genome-wide estimates were subsequently not conducted.

The estimated species divergence time of $t$ = 38.7–47.0 mya (XLA.S-XLA.L, range among chromosomes) and $t$ = 44.8–64.3 mya (XTR-XLA) (S2 Table) is older than the previous estimates (using genomic data) of 34 and 48 mya, respectively [6], but similar to some other studies. For example, based on mitochondrial DNA, the common ancestor of *Xenopus* frogs was dated back to 31.8–54.6 mya [24]. In addition, the divergence time between *Xenopus* and *Silurana* (based on DNA, morphology, and fossil calibration) was estimated to be 27–51 mya [25]. Furthermore, simple molecular clock estimates date the *X. laevis–X. tropicalis* divergence and genome duplication in *X laevis* at around 50 and 40 mya, respectively [26].

The slight discrepancies in $t$ could be attributed to the past assumption that divergence of sequence is equivalent to species divergence time, but actually $N_A$ can influence divergence time of sequences. Our study using a maximum likelihood approach has accounted for the impact of potentially large $N_A$ on estimations of divergence time and substitution rate. From the ML estimates, we also calculated an ancestral population size of $N_A$ = 0.88–2.58 x $10^6$ (XLA.S-XLA.L, range among chromosomes) and $N_A$ = 0.96–4.02 x $10^6$ (XTR-XLA) (S2 Table). This is an unexpectedly large estimated population size, considering that it is comparable to that of fruitflies ($N_e$ = 2–7 x $10^6$), and at least one order of magnitude higher than lizards and bony fish [27]. In extant humans, the estimated effective population is much lower at below $10^4$ [28, 29].

Even though naturally-occurring species hybridization is restricted to a few *Xenopus* spp. (including *X. laevis* and excluding *X. tropicalis*) [30, 31], it seems possible that population structure and local hybridization may have contributed to a large estimated population size. Indeed, the hybridization of *X. laevis* (L) and (S) genomes as well as the fusion of chromosomes 9 and 10 and various intrachromosomal recombinations may have contributed to high population size estimations. Recently, there is emerging evidence supporting that hybridization may have an important role in adaptation [32–34].

The seemingly large population size is based on our large ML estimates of *X*, which is dependent on both population size and mutation rate. Therefore, high mutation rate may be contributive to the large *X* estimate. We briefly examined variation in *μ* using the ML estimates of $Y = 2\mu t$ and constant species divergence time (48 mya and 34 mya for divergence times of *X. laevis*—*X. tropicalis* and *X. laevis* L–*X. laevis* S, respectively [6]) and found that *μ* varied between chromosomes (1.85–2.59 and 2.10–2.59 x$10^{-9}$ substitutions/site/year, respectively). However, no clear increase was evident, supporting that large population size instead of mutation rate more likely contributed to the large ML estimates of *X*. Future investigation of genomic sequences of other *Xenopus* spp. as well as *X. tropicalis* or *X. laevis* sub-populations will aid in elucidating the mechanisms behind these large estimations. Indeed, the *Xenopus* topology and taxonomy validated using morphological and molecular analyses (mitochondrial and nuclear genes) [5], with a large diversity of polyploidy and high number of independent polyploidization events, could be further explored using genomics. Moreover, genomic studies of other species within the sub-genus *Silurana*, including *X. mellotropicalis* and *X. epitropicalis* [35], will be important to further understand the relationship between chromosomal rearrangements and mutation rates before and after polyploidization.

## Conclusions

Using available genome data, we have demonstrated the presence of synonymous substitution rate heterogeneity within *Xenopus* frogs at the inter- and intra-chromosomal level. We found that chromosome 10 had higher *k* compared to other chromosomes, as well as peculiar intrachromosomal patterns, likely related to the fusion of chromosomes 9 and 10 in *X. laevis*. Within most other chromosomes, we identified a pattern of higher *k* at both the distal and proximal ends of the chromosomes, and this may be caused by more frequent recombination and elevated mutation rates in these regions. In addition, maximum likelihood estimations provided additional evidence of rate heterogeneity across all chromosomes. Our estimated species divergence times are a little different to that of a previous study, possible because our approach has accounted for ancestral population size. Our study is one of the first to estimate ancestral population size in *Xenopus* frogs, which we found to be surprisingly high. This large population size may be attributed to hybridization and population structure and thus warrants further investigation to validate our findings.

## Supporting information

**S1 Fig. Relationship between proportion of synonymous substitutions (*k*) and location within each chromosome, for each genome pair combination (XTR-XLA.L in red, XTR-XLA.S in green, and XLA.L-XLA.S in blue).** Representative plots are presented in Fig 2. Centromere locations (vertical dotted line) are based on [15, 16].
(DOCX)

**S2 Fig. Frequency of CG dinucleotides along each chromosome based on whole genome data of *Xenopus tropicalis* and *X. laevis* L and S.** Y-axes represent CG composition in 1Mb

sliding windows with a 100kb step on the chromosomes, X-axes represent chromosome locations (x1000000). The dinucleotide composition was computed using a batch-learning self-organizing map (BLSOM) program (Abe et al. 2003). The BLSOM program can be obtained from UNTROD, Inc. (y_wada@nagahama-i-bio.ac.jp).
(DOCX)

**S3 Fig.** Unique patterns observed between proportion of synonymous substitutions *k* and location within (A) chromosome 8, (B) chromosome 3 (normal pattern), and (C) chromosome 9/10 could be attributed to chromosomal rearrangements previously identified using cytogenetic mapping by Session et al. [5].
(DOCX)

**S1 Table. Quadratic regression analyses of relationship between W = chromosome location and Z = proportion of synonymous substitutions (*k*).** Partial F-test was used to compare two models: linear [lm(*W~Z*)] and quadratic [lm(*W~Z + I (Z^2)*)]. Values presented are significance probabilities associated with the F values, Pr(F), and were calculated independently for each pairwise comparison of genomes: XLA.L-XLA.S (blue), XTR-XLA.L (red), and XTR-XLA.S (green). Values with *Pr(F)* < 0.01 (in bold) significantly support a quadratic model.
(DOCX)

**S2 Table. Maximum likelihood estimates and calculations of rate heterogeneity (α), ancestral population size (NA), and divergence time (t).** Although chromosome 9_10 is homoeologous in XLA.L and XLA.S, we separated estimations to chromosome 9 or 10 based on location in XTR.
(DOCX)

**S3 Table. Chi-square p-values following fitting observed data (synonymous difference) against the distribution 'nbinom' by maximum likelihood.**
(DOCX)

## Acknowledgments

We thank Akira Sasaki for assistance with statistical analyses, and Naoyuki Takahata for critical feedback on the manuscript. We acknowledge the Xenopus Community in Japan (XCIJ) and National Bio-Resource Project (NBRP) for the XenOrtho database.

## Author Contributions

**Conceptualization:** Quintin Lau, Takeshi Igawa, Yoko Satta.

**Formal analysis:** Quintin Lau, Takeshi Igawa, Yoko Satta.

**Funding acquisition:** Hajime Ogino, Yoko Satta.

**Investigation:** Quintin Lau, Takeshi Igawa, Yukako Katsura, Toshimichi Ikemura.

**Supervision:** Hajime Ogino, Yoko Satta.

**Writing – original draft:** Quintin Lau.

**Writing – review & editing:** Takeshi Igawa, Hajime Ogino, Yukako Katsura, Toshimichi Ikemura, Yoko Satta.

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
