## [Decision Letter · Decision Letter 0]

25 May 2020

PONE-D-20-12594

Heterogeneity of synonymous substitution rates in the Xenopus frog genome

PLOS ONE

Dear Dr. Lau,

Thank you for submitting your manuscript to PLOS ONE. After careful consideration, we feel that it has merit but does not fully meet PLOS ONE’s publication criteria as it currently stands. Therefore, we invite you to submit a revised version of the manuscript that addresses the points raised during the review process.

Both reviewers raise important points about the writing, consistency, and references.

We look forward to receiving your revised manuscript.

Kind regards,

Marc Robinson-Rechavi

Academic Editor

PLOS ONE

Journal Requirements:

Reviewers' comments:

Reviewer's Responses to Questions

**Comments to the Author**

1. Is the manuscript technically sound, and do the data support the conclusions?

Reviewer #1: Partly

Reviewer #2: Partly

2. Has the statistical analysis been performed appropriately and rigorously? 

Reviewer #1: Yes

Reviewer #2: I Don't Know

3. Have the authors made all data underlying the findings in their manuscript fully available?

Reviewer #1: Yes

Reviewer #2: Yes

4. Is the manuscript presented in an intelligible fashion and written in standard English?

Reviewer #1: Yes

Reviewer #2: Yes

5. Review Comments to the Author

Reviewer #1: The proposed manuscript shows important piece of work and supplements Xenopus phylogeny.

1) I suggest some improvements of the discussion section which seems to be little bit poor. Authors are mainly focused on the study Session et al. (2016) but there are a lot of other important phylogenetic studies which authors should take into account and discussed. e.g., Evans et al. (2004) estimated divergent times including common ancestor of S. tropicalis and X. laevis. Cannatella (2015) estimated divergence times based on fossil calibration. New re-validation of the Xenopus topology and taxonomy was done by Evans et al. (2015) including downgrading "Silurana" from a genus to a subgenus within the genus "Xenopus". Then Silurana allotetraploid evolution was confirmed cytogenetically by Knytl et al (2017).

2) I do not understand why there are three separate values for chromosome 9, 10 and 9_10 - the column "proportion of synonymous substitutions XLA.L-XLA.S" (Table 2).

3) Authors use terms "homoeologous" and "homeologous". It should be uniform.

4) Higher substitution rate was found at proximal and distal ends of chromosomes caused probably by frequent recombination and mutations. Are there any studies focused on a presence of transpositions in these parts of chromosomes? If yes, it might be discussed.

5) There are identical labels of second and third column in the Figure 3A and B (XLA.L-XTR).

Cannatella D. (2015). Xenopus in space and time: fossils, node calibrations, tip-dating, and paleobiogeography. Cytogenetic and genome research, 145(3-4), 283-301.

Evans BJ, Carter TF, Greenbaum E, Gvoždík V, Kelley DB, McLaughlin PJ, et al. (2015) Genetics, Morphology, Advertisement Calls, and Historical Records Distinguish Six New Polyploid Species of African Clawed Frog (Xenopus, Pipidae) from West

and Central Africa. PLoS ONE 10(12): e0142823.

Evans BJ, Kelley DB, Tinsley RC, Melnick DJ, & Cannatella DC. (2004). A mitochondrial DNA phylogeny of African clawed frogs: phylogeography and implications for polyploid evolution. Molecular phylogenetics and evolution, 33(1), 197-213.

Knytl M, Smolík O, Kubíčková S, Tlapáková T, Evans BJ, Krylov V (2017) Chromosome divergence during evolution of the tetraploid clawed frogs, Xenopus mellotropicalis and Xenopus epitropicalis as revealed by Zoo-FISH. PLoS ONE 12(5): e0177087.

Session AM, Uno Y, Kwon T, Chapman JA, Toyoda A, Takahashi S, et al. (2016). Genome evolution in the allotetraploid frog Xenopus laevis. Nature, 538(7625), 336-343.

Reviewer #2: The manuscript “Heterogeneity of synonymous substitution rates in the Xenopus frog genome”, by Lau et al provides an important assessment of population parameters in a model system that has become critical to understanding the impacts of genome duplication on genome structure and evolution.

I have only a few comments that I think should be addressed prior to publication. These are listed below.

1) Line 169. Is the word “somehow” necessary? It seems to indicate that there is some doubt. I would recommend expanding on this or removing “somehow”

2) Line 217 The statement “While amphibians do not have microchromosomes” is false. Many taxa have microchromsomes and it is abundantly clear that they were present in the common ancestor of all amphibians. Xenopus do not have microchromosomes however.

3) Line 230, there seems to be a reference to a non-existent figure.

4) In the discussion of apparent increases in population size. Is it possible that this is accounted for to some degree by increases/changes in mutation rate. I’d recommend including more detail on how the relative contributions of Ne and mu were assessed.

5) Lines 254-259. The proposal on ongoing hybridization and large population size seems to run counter to the idea that the two founding populations had accumulated distinctly different repetitive element contents (from the X laevis genome paper PMID: 27762356) as large Ne would seemingly tend to work against fixation of large numbers of semi-deleterious insertions and hybridization would tend to homogenize repeat family contents. Perhaps a bit more discussion would make it easier to integrate the apparent conflicts.

Sincerely

Jeramiah Smith

6. PLOS authors have the option to publish the peer review history of their article (what does this mean?). If published, this will include your full peer review and any attached files.

Reviewer #1: No

Reviewer #2: Yes: Jeramiah James Smith

---

## [Author Response · Author response to Decision Letter 0]

15 Jun 2020

Dear Prof. Robinson-Rechavi,

Thank you for facilitating the review process of our manuscript. We also thank both reviewers for their insightful comments and feedback that have helped us to improve our manuscript. We have addressed all reviewer comments below, and have amended the manuscript accordingly. All line references refer to our 'Revised Manuscript with Track Changes'. References refer to those listed in the revised manuscript.

We hope that you and the reviewers find our amendments to be satisfactory.

Regards

Quintin Lau

Reviewer #1: The proposed manuscript shows important piece of work and supplements Xenopus phylogeny.

1) I suggest some improvements of the discussion section which seems to be little bit poor. Authors are mainly focused on the study Session et al. (2016) but there are a lot of other important phylogenetic studies which authors should take into account and discussed. e.g., Evans et al. (2004) estimated divergent times including common ancestor of S. tropicalis and X. laevis. Cannatella (2015) estimated divergence times based on fossil calibration. New re-validation of the Xenopus topology and taxonomy was done by Evans et al. (2015) including downgrading "Silurana" from a genus to a subgenus within the genus "Xenopus". Then Silurana allotetraploid evolution was confirmed cytogenetically by Knytl et al (2017).

Thank you for your recommendation to improve the Discussion section and providing of several important references. We have now incorporated all suggestions into the discussion (and partially in the introduction):

Line 55-57: “Indeed, revalidation of Xenopus topology and taxonomy [5] classifies Silurana as a subgenus within Xenopus, which includes X. tropicalis.”

Line 254-261: “The estimated species divergence time of t = 38.7 – 47.0 mya (XLA.S-XLA.L, range among chromosomes) and t = 44.8 - 64.3 mya (XTR-XLA) (Table S2), is older than the previous estimates (using genomic data) of 34 and 48 mya, respectively [6], but similar to some other studies. For example, based on mitochondrial DNA, the common ancestor of Xenopus frogs was dated back to 31.8 – 54.6 mya [24]. In addition, the divergence time between Xenopus and Silurana (based on DNA, morphology, and fossil calibration) was estimated to be 27 – 51 mya [25]. Furthermore, simple molecular clock estimates date the X. laevis – X. tropicalis divergence and genome duplication in X laevis at around 50 and 40 mya, respectively [26].”

Line 291-294: “Moreover, genomic studies of polyploid species within the sub-genus Silurana, including X. mellotropicalis and X. epitropicalis [35], will be important to further understand the relationship between chromosomal rearrangements and mutation rates before and after polyploidization.

2) I do not understand why there are three separate values for chromosome 9, 10 and 9_10 - the column "proportion of synonymous substitutions XLA.L-XLA.S" (Table 2).

To avoid confusion, we have now deleted the ‘proportion of synonymous substitutions’ data for chromosomes 9 and 10 in the ‘XLA.L-XLA.S’ column, and the corresponding footnote.

3) Authors use terms "homoeologous" and "homeologous". It should be uniform.

We have ensured that the term is now used uniformly in the manuscript (see line 196).

4) Higher substitution rate was found at proximal and distal ends of chromosomes caused probably by frequent recombination and mutations. Are there any studies focused on a presence of transpositions in these parts of chromosomes? If yes, it might be discussed.

Thank you for your suggestion. Transposable elements have been studied in Xenopus frogs, but not in association with specific parts of the chromosomes, but instead with GC content. A study in Drosophila found higher frequency of repeat elements near the center of autosomes. We have now included the following sentences into lines 174-184:

“While an association between GC content and recombination rate has not been demonstrated in amphibians to date, a positive correlation has been identified in many other species [15–17]. The presence of transposable elements (TEs) may also contribute to heterogeneity in substitution rates within a chromosome. In Drosophila melanogaster, TEs tended to be accumulated in the proximal region (i.e. near centromere) of autosomes [18]; transposon density was also found to be negatively correlated with recombination rate in this species [19]. Moreover, a study in X. tropicalis found that frequency of TEs was negatively correlated with GC content [20], but chromosomal location of only specific transposons have been mapped in X. laevis [6]. Based on these previous studies, there may be a lower frequency of TEs in the distal parts of the Xenopus chromosomes along with high CG content, which may be related to high recombination rates.”

5) There are identical labels of second and third column in the Figure 3A and B (XLA.L-XTR).

Sorry for our error and thank you for spotting it. We have now changed the labels to be XTR-XLA.L and XTR-XLA.S for the second and third columns, respectively.

Reviewer #2: The manuscript “Heterogeneity of synonymous substitution rates in the Xenopus frog genome”, by Lau et al provides an important assessment of population parameters in a model system that has become critical to understanding the impacts of genome duplication on genome structure and evolution.

I have only a few comments that I think should be addressed prior to publication. These are listed below.

1) Line 169. Is the word “somehow” necessary? It seems to indicate that there is some doubt. I would recommend expanding on this or removing “somehow”

As suggested, we have now removed ‘somehow’ to avoid ambiguity.

2) Line 217 The statement “While amphibians do not have microchromosomes” is false. Many taxa have microchromsomes and it is abundantly clear that they were present in the common ancestor of all amphibians. Xenopus do not have microchromosomes however.

I apologize for this error, and have amended the statement to ‘While Xenopus frogs do not have microchoromosomes’. (Line 229)

3) Line 230, there seems to be a reference to a non-existent figure.

I’m sorry that I do not quite follow this query. Line 230 (now line 242) is the Figure title for Figure 3, and line 243-247 is the Figure 3 caption. I have double-checked that Figure 3 was attached in the initial uploaded manuscript. 

4) In the discussion of apparent increases in population size. Is it possible that this is accounted for to some degree by increases/changes in mutation rate. I’d recommend including more detail on how the relative contributions of Ne and mu were assessed.

Due to the nature of our maximum likelihood estimates of X=4N_A gμ and Y=2μt [N_A, ancestral population size; t, divergence time; g, generation time; and µ, mutation rate (synonymous substitution rate) per site per year], assumptions in one of the parameters had to be made. In our manuscript, a constant µ was assumed based on genome analysis in a previous study. We did examine ML estimates of µ after assuming T is 48 mya and 34 mya for divergence between X. laevis - X. tropicalis and X.laevis L – X.laevis S, respectively. It may be possible that the apparently high population size could be due to variation or increase in mutation rate. Therefore, we have added the following statement in lines 278–291:

“The seemingly large population size is based on our large ML estimates of X, which is dependent on both population size and mutation rate. Therefore, high mutation rate may be contributive to the large X estimate. We briefly examined variation in μ using the ML estimates of Y=2μt and constant species divergence time (48 mya and 34 mya for divergence times of X. laevis - X. tropicalis and X. laevis L – X. laevis S, respectively [6]) and found that µ varied between chromosomes (1.85 – 2.59 and 2.10 – 2.59 x10-9 substitutions/site/year, respectively). However, no clear increase was evident, supporting that large population size instead of mutation rate more likely contributed to the large ML estimates of X. Indeed, the Xenopus topology and taxonomy validated using morphological and molecular analyses (mitochondrial and nuclear genes) [5], with a large diversity of polyploidy and high number of independent polyploidization events, could be further explored using genomics.”

5) Lines 254-259. The proposal on ongoing hybridization and large population size seems to run counter to the idea that the two founding populations had accumulated distinctly different repetitive element contents (from the X laevis genome paper PMID: 27762356) as large Ne would seemingly tend to work against fixation of large numbers of semi-deleterious insertions and hybridization would tend to homogenize repeat family contents. Perhaps a bit more discussion would make it easier to integrate the apparent conflicts.

Thank you for your comment here. Perhaps there may have been some misunderstanding, as we feel that there is no conflict between hybridization and large population size.

After the divergence of X. laevis L and S progenitors, a number of transposable elements were preserved in the original state, according to identification of sub-genome specific TEs (Session et al. 2016; PMID: 27762356). This instead supports that repeat family elements were not homogenized in the genome after the hybridization between the X. laevis L and S progenitors. Therefore, we feel that ‘hybridization would tend to homogenize repeat family contents’ may not be applicable here. Moreover, many genes have been retained as duplicates following whole genome duplication (e.g. Chain et al 2011; PMID: 22151890), while others like MHC have copy number loss (Sato et al 1994; PMID: 8454860). 

Hybridization can increase the heterogeneity of a species (i.e. two species joining to become one). Consequently, genetic variation within a species should increase, and large genetic variation is reflected in a large effective population size. Based on this, we feel that there are no apparent conflicts. However, we are happy to discuss further with you in case our discussion is missing some clarity.

---

## [Decision Letter · Decision Letter 1]

9 Jul 2020

Heterogeneity of synonymous substitution rates in the Xenopus frog genome

PONE-D-20-12594R1

Dear Dr. Lau,

We’re pleased to inform you that your manuscript has been judged scientifically suitable for publication and will be formally accepted for publication once it meets all outstanding technical requirements.

Kind regards,

Marc Robinson-Rechavi

Academic Editor

PLOS ONE

Additional Editor Comments (optional):

Reviewers' comments:

Reviewer's Responses to Questions

**Comments to the Author**

1. If the authors have adequately addressed your comments raised in a previous round of review and you feel that this manuscript is now acceptable for publication, you may indicate that here to bypass the “Comments to the Author” section, enter your conflict of interest statement in the “Confidential to Editor” section, and submit your "Accept" recommendation.

Reviewer #1: All comments have been addressed

Reviewer #2: (No Response)

2. Is the manuscript technically sound, and do the data support the conclusions?

Reviewer #1: Yes

Reviewer #2: Partly

3. Has the statistical analysis been performed appropriately and rigorously? 

Reviewer #1: I Don't Know

Reviewer #2: Yes

4. Have the authors made all data underlying the findings in their manuscript fully available?

Reviewer #1: Yes

Reviewer #2: Yes

5. Is the manuscript presented in an intelligible fashion and written in standard English?

Reviewer #1: Yes

Reviewer #2: Yes

6. Review Comments to the Author

Reviewer #1: The proposed manuscript was improved and I did not find any errors or discrepancies. My comments and suggestions were adequately addressed as well. Only one last point, I am not a statistical expert and for this reason I suggest that someone else as an expert should briefly check statistical data.

Reviewer #2: The manuscript is certainly improves, although the results related to the interrelationship between population size estimates and hybridization are a bit difficult to digest and it seems that these would be much easier to interpret if a figure were added to the discussion that annotates changes in population size/divergence times, and where hybridization and other factors may impact their estimates. Are the hybridization events alluded to here independent of the allopolypolidization event, or does this potentially impact those estimates?

7. PLOS authors have the option to publish the peer review history of their article (what does this mean?). If published, this will include your full peer review and any attached files.

Reviewer #1: **Yes: **Martin Knytl

Reviewer #2: **Yes: **Jeramiah Smith